# Functional Conservation and Divergence of Five *AP1/FUL*-like Genes in Marigold (*Tagetes erecta* L.)

**DOI:** 10.3390/genes12122011

**Published:** 2021-12-17

**Authors:** Chunling Zhang, Yalin Sun, Xiaomin Yu, Hang Li, Manzhu Bao, Yanhong He

**Affiliations:** 1Key Laboratory of Horticultural Plant Biology, Ministry of Education, College of Horticulture and Forestry Sciences, Huazhong Agricultural University, Shizishan Street No. 1, Wuhan 430070, China; zhangchunling@webmail.hzau.edu.cn (C.Z.); Yxm.223@webmail.hzau.edu.cn (X.Y.); hangli@webmail.hzau.edu.cn (H.L.); mzbao@mail.hzau.edu.cn (M.B.); 2Institute of Vegetable, Wuhan Academy of Agricultural Sciences, Wuhan 430207, China; kevincan@163.com

**Keywords:** *Tagetes erecta*, MADS-box genes, *APETALA1/FRUITFULL*, yeast two-hybrid, gene function

## Abstract

Members of *AP1/FUL* subfamily genes play an essential role in the regulation of floral meristem transition, floral organ identity, and fruit ripping. At present, there have been insufficient studies to explain the function of the *AP1/FUL*-like subfamily genes in Asteraceae. Here, we cloned two *euAP1* clade genes *TeAP1-1* and *TeAP1-2*, and three *euFUL* clade genes *TeFUL1*, *TeFUL2*, and *TeFUL3* from marigold (*Tagetes erecta* L.). Expression profile analysis demonstrated that *TeAP1-1* and *TeAP1-2* were mainly expressed in receptacles, sepals, petals, and ovules. *TeFUL1* and *TeFUL3* were expressed in flower buds, stems, and leaves, as well as reproductive tissues, while *TeFUL2* was mainly expressed in flower buds and vegetative tissues. Overexpression of *TeAP1-2* or *TeFUL2* in Arabidopsis resulted in early flowering, implying that these two genes might regulate the floral transition. Yeast two-hybrid analysis indicated that TeAP1/FUL proteins only interacted with TeSEP proteins to form heterodimers and that TeFUL2 could also form a homodimer. In general, TeAP1-1 and TeAP1-2 might play a conserved role in regulating sepal and petal identity, similar to the functions of MADS-box class A genes, while *TeFUL* genes might display divergent functions. This study provides a theoretical basis for the study of *AP1/FUL*-like genes in Asteraceae species.

## 1. Introduction

Although flowers show great diversity in morphology, structure, composition, color, and function, they are usually composed of four distinct concentric whorl floral organs: sepals in the outermost whorl, petals in the second whorl, stamens (male reproductive organs) in the third whorl, and carpels (female reproductive organs) in the innermost whorl [1]. The exploration of the mechanism of distinct floral organ formation has undergone a long-term challenge in plant developmental genetics [1,2,3,4]. Based on the study of the floral formation in model plants, such as *Arabidopsis thaliana* L., *Antirrhinum majus* L., and *Petunia hybrida* Vilmorin, the fate of the different floral organs was considered to be determined by a complex regulatory network composed of MADS-box proteins [5].

The MADS-box protein family is one of the most widely studied transcriptional factor families in angiosperm, and this family plays a key role in regulating floral meristem development, floral organ identity, fruit and seed development, vegetative tissue development, and flowering time [6,7,8]. MADS box genes in higher plants are reported to have undergone several duplication events that promote the evolution of morphological complexity of the flower [9], thus allowing MADS-box genes to cluster into several major subfamilies [10,11,12]. One subfamily of MADS-box genes forms the angiosperm-specific *APETALA1/FRUITFUL* (*AP1/FUL*) lineage via gene duplication. Phylogenetic analyses reveal that *AP1/FUL* lineage has undergone numerous duplication events throughout angiosperm diversification [13,14]. *AP1* genes diverge into two types of *AP1* lineage genes within the core eudicot, namely, *euAP1* (Arabidopsis *AP1* and Antirrhinum *SQUA*) and *euFUL*, which is likely to be part of the whole genome duplication event before the diversification of the core eudicots, and this event is often known as the γ event [15,16]. Contrary to core eudicots, non-core eudicots have only *FUL*-like clade genes [13,14,17,18]. Within the model plant Arabidopsis, the *euAP1* clade is subdivided into *AP1* and *CAULIFLOWE* (*CAL*) genes [19,20], whose amino acid sequences possess an acidic domain and a farnesylation motif (CaaX) in their 3′ end of the coding sequence [13,21]. *AP1* and *CAL* are accumulated in floral meristems, sepals, and petals primordia [22,23]. In Arabidopsis, *ap1* mutation results in the absence of petals, the transformation of sepals into bract-like structures, and the production of secondary flowers from the axils of the first whorl organs [22,24]. Overexpression of *AP1* leads to remarkable early flowering and transformation of inflorescence shoot apical meristem into floral meristem [25]. In single Arabidopsis *cal* mutants, there are no remarkable changes in floral organs, but the *cal* mutation enhances the repetitive branching pattern in the floral meristem of *ap1* mutants [24,26]. In other core eudicots, the *ap1* mutation only changes the sepal structure, but it does not affect petal structure [27,28]. Furthermore, ectopic expression of *Fortunella crassifolia* Swingle *FcAP1* in Arabidopsis [29] and heterologous overexpression of the *AP1*-like gene *Betula pendula* Roth. *BpMADS3* [30] and *Pisum sativum* L. *PEAM4* [31] in *Nicotiana tabacum* L. cause early flowering with floral meristem development unaffected. In addition, *AP1*, together with *TFL1* and *LFY*, is the key inflorescence regulator in Arabidopsis [22,32,33]. In general, *euAP1* clade genes exhibit a conserved function in specifying the floral meristem and sepal identity in core eudicots.

The *euFUL* genes show a conserved function in promoting the transition from vegetative meristems to reproductive meristems and in regulating fruit development in core eudicots. The euFUL proteins are characterized by possessing a conserved six-hydrophobic-amino-acid motif (FUL-like motif) in the C domain [13,14]. This motif is considered to be conserved in the entire ancestral gene lineage, and its occurrence is prior to the *euFUL*/*euAP1* duplication, but its function remains unclear [13,14]. In Arabidopsis, *euFUL* is divided into *FUL* (or *euFUL**I*) and *AGAMOUS*-like *79* (*AGL79* or *euFULII*) in the duplication event of *euFUL* clade [34,35,36]. Different from *euAP1*, *FUL* is mainly expressed in growing leaves, inflorescence meristems, carpel primordia, and young siliques [25,37]. Arabidopsis *FUL* is redundant with *AP1* and *CAL* in regulating the floral meristem identity, and it also regulates the flowering time, axillary meristem activation, meristem determinacy, and plant longevity [23,38]. In *ful* mutants, the cauline leaf development was terminated, and the floral development was disrupted [37,39]. The transcript of *AGL79* is detected in roots, but its function remains unclear [34,35]. Recently, only the limited functional analysis of *euFUL* genes in other core eudicots is available. Similar to the *FUL* gene function in Arabidopsis, ectopic expression of *DEFICIENS-homolog28* (*DEFH28*, a *euFULII* gene, Antirrhinum) [40] in Arabidopsis causes early flowering, transformation of inflorescence into a terminal flower, and silique indehiscence. Moreover, overexpression of *NtFUL* in *N. tabacum* also results in early flowering and failure in capsule dehiscence [41]. In *P. hybrida*, silencing *PETUNIA FLOWERING GENE* (*PFG*, an *euFULⅠ* gene) leads to the interruption of inflorescence formation, thus maintaining vegetative growth [42].

Asteraceae is one of the most abundant and widespread families of flowering plants, and it has a specific capitulum consisting of two flower types: the outer are the sterile ray florets and the inner are the fertile disk florets. The specific inflorescence makes Asteraceae a suitable material for studying the evolution and function of MADS-box genes related to floral organ development. Nevertheless, the available functional information on *AP1/FUL* genes in Asteraceae is restricted to *Gerbera hybrida* Hort. [43], *Chrysanthemum lavandulifolium* (Fisch. ex Trautv.) Ling et Shih [44], and *Chrysanthemum morifolium* Ramat. [45,46]. The expression patterns of *AP1/FUL*-like genes vary with various Asteraceae species. The *euAP1*-like genes *CDM111 (C*. *morifolium*) and *GASQUA3* (*G**. hybrida*) are highly expressed in sepals and petals, while their homologous gene *GASQUA1* (*G**. hybrida*) is not expressed in floral meristem or in perianth primordia [43,45,47]. The *FUL*-like genes *GSQUA2* and *GSQUA5* are only expressed in florescence and floral organs, while *GSUA4* and *CDM41* are also expressed in leaves [43,45]. The function of *AP1/FUL*-like genes in Asteraceae remains unclear. Overexpression of the *AP1*-like gene *CDM111* or *FUL*-like gene *ClM8* in Arabidopsis results in the altered flowering time and inflorescence structure [44,45]. However, overexpression of *GSQUA2* (a *FUL*-like gene, homolog to *DEFH28*) in gerbera results in a dwarf plant, early flowering, and vegetative abnormality, but it does not affect inflorescence structure [43].

Marigold (*T. erecta* ) is a popular ornamental plant and economic crop, whose flowers are rich in lutein. As a member of Asteraceae, marigold also has a typical capitulum. Compared with the complex inflorescence structure of *G. hybrida*, marigold wears a simple inflorescence, which consists of two distinct flower types, namely, the outermost ray flowers and the disk flowers. Furthermore, the ray flowers in marigold retain female pistils, whereas the ray flowers in *Helianthus annuus* L. are sterile because of only filamentous remnants in aborted stamens and empty ovaries. In addition, the whole life cycle of marigold lasts only 2-3 months from sowing to flowering. Furthermore, in the evolutionary history of the Asteraceae family, marigold undergoes a long evolution process and it belongs to the Tageteae clade [48]. These characteristics make marigold a valuable material for studying the molecular mechanism of marigold inflorescence formation. In our previous work, we have obtained functionally characterized class B (TePI, TeAP3-1, TeAP3-2, TeTM6-1, and TeTM6-2) [49], C (TeAG1 and TeAG2) [50], D (TeAGL11-1 and TeAGL11-2) [50], and E (TeSEP1, TeSEP3-1, TeSEP3-2, TeSEP3-3, and TeSEP4) [51] genes, which are active during marigold inflorescence and floret development and have specific expression patterns in floral organs. In this study, we cloned and characterized five *AP1/FUL*-like genes in marigold, whose distinct expression patterns, protein interaction patterns, and different phenotypes in Arabidopsis transgenic lines might imply divergent functions of these five genes in regulating the floral meristem development, floral organ identity, and flowering time.

## 2. Materials and Methods

### 2.1. Plant Materials and Growth Conditions

Marigold (*T. erecta*, M525B-1) is an inbred line with more than 10 generations of continuous self-crossing, and it has only one whorl of ray florets outside the capitulum [52]. Marigold plants were grown in the experimental field of Huazhong Agricultural University (lat. 30°28’36.5” N, long, 114°21’59.4” E) under natural conditions. To investigate *AP1/FUL*-like genes expression patterns, the samples of vegetative tissues, flower buds from different stages, and floral organs in the blooming period were collected as described by Ai et al. [49], and were frozen immediately in liquid nitrogen and stored at −80 °C.

Arabidopsis ecotype Columbia (Col-0) plants were used for functional analysis of *AP1*/*FUL*-like genes of marigold. Plants were grown in a chamber at 22 °C under long-day conditions (16 h light, 8 h dark) with 70% relative humidity.

### 2.2. Total RNA Extraction, Isolation, and Bioinformatics Analysis of AP1/FUL-like Genes from Marigold

The total RNA of each sample was isolated with a PLANTpure kit (Aidlab, Beijing, China) according to the manufacturer’s protocol. The quantity and the quality of RNA samples were analyzed by a Nano-Drop 2000 Spectrophotometer (Thermo Fisher Scientific, Wilmington, DE, USA) and by running test gels with ethidium bromide staining. The first-strand cDNA was synthesized by the TRUEscript RT reagent Kit with gDNA Eraser (Aidlab, Beijing, China) with the Oligo-dT primers. Five *AP1/**FUL*-like genes were selected from the transcriptomic data (accession number SRP066084) [49] and full-length transcriptomic data (unpublished), and named *TeAP1-1*, *TeAP1-2*, *TeFUL1*, *TeFUL2**,* and *TeFUL3*, respectively. To verify the accuracy of these five gene sequences, the specific primers *TeAP1-1*-full-F/R, *TeAP1-2*-full-F/R, *TeFUL1*-full-F/R, *TeFUL2*-full-F/R, and *TeFUL3*-full-F/R were designed in the 3’ and 5’ terminal region by Primer Premier 5 software (Premier Biosoft International, Palo Alto, CA, USA) (Appendix A) and used to clone full-length gene sequences. The PCR was programmed using the following parameters: 94 °C for 4 min; 38 cycles of 94 °C for 30 s, 56 °C for 30 s, and 72 °C for 2 min, and a final extension at 72 °C for 10 min. The PCR amplification fragments were purified and then cloned into a *pMD18-T* vector (Takara, Dalian, China). Positive clones were verified by PCR using *M13*-F/R universal primers, and 3–5 positive clones were selected and sequenced by the Sangon company in Shanghai.

The Open Reading Frames (ORFs) of these five *AP1/FUL*-like genes were predicted online (https://www.ncbi.nlm.nih.gov/orffinder/ access on 2 November 2021) and were blasted against the NCBI to search for their homologous sequences. To identify the conserved motifs of AP1/FUL amino acids, the multiple sequence alignment was performed by using the DNAMAN (v.6.0) software (https://www.lynnon.com access on 2 November 2021) and BoxShade (https://embnet.vital-it.ch/software/BOX_form.html access on 2 November 2021). A total of 36 *AP1/FUL*-like genes and 4 *AGL6*-like genes derived from model plants and *G. hybrida* were downloaded from the National Center for Biotechnology Information (NCBI) (http://www.ncbi.nlm.nih.gov access on 2 November 2021) for phylogenetic analysis. The gene accession number was listed in Appendix A. Four AGL6-like proteins from model plants (Arabidopsis and petunia) and Asteraceae species (gerbera) were used as the outgroup. The construction of the phylogenetic tree was based on amino acid alignment with the default settings of MUSCLE in MEGA (v. 7.0). A phylogenetic tree was constructed by the neighbor-joining (NJ) method with bootstrap confidence values of 1000 replicates, and distances were calculated with Poisson corrections for multiple substitutions.

### 2.3. Gene Expression Analysis

To analyze the expression of five A class genes in marigold, the total RNA from the samples of roots, tender stems, fresh leaves, different sizes of flower buds (0–1, 2–3, 4–5 and 6–7 mm in diameter, respectively), sepals, petals and pistils of ray and disk florets, stamens of disk florets, receptacles, bracts, and ovaries of opened flowers were isolated with PLANTpure kit (Aidlab, Beijing, China) according to the manufacturer’s protocol. The total RNA was reverse-transcribed by the TRUEscript RT reagent Kit with gDNA Eraser (Aidlab, Beijing, China) with the Oli-go-dT primers. The reverse transcription reaction contained total RNA 1 μg, 4 × gDNA Eraser mix 4 μL, 5 × TRUE RT MasterMixⅡ (Aidlab, Beijing, China), and double-distilled water to supply a final volume of 20 μL. The reverse transcription reactions were incubated at 42 °C for 20 min and 85 °C for 5 s. The analysis of expression patterns of *AP1/FUL*-like genes in different tissues and different development stages of flower buds was performed by quantitative real-time PCR (qRT-PCR). The specific primers were designed within the non-conservative C-terminal region using the Primer Premier 5.0 software to amplify products between 90 and 200 bp (Appendix A), and the products were named *qTeAP1-1*, *qTeAP1-2*, *qTeFUL1*, *qTeFUL2*, and *qTeFUL3*. The specific and unique PCR products for each primer pair were confirmed by 1.2% agarose gel electrophoresis. The qRT-PCR was carried out in an optical 384-well plate in the QuantStudio 6 Flex real-time PCR system (Applied Biosystems, Palo Alto, CA, USA) with SYBR Primix Ex Taq kit (TaKaRa, Dalian, China) according to manufacturer’s instructions. The qRT-PCR data were analyzed in the ABI 7500 Detection System (Applied Biosystems, Palo Alto, CA, USA). The qRT-PCR products were amplified in 10 μL reaction solution containing 1 μL template of the reaction mixture, 5 μL 2 × SYBR Green Master Mix (TaKaRa, Dalian, China), 0.2 μL forward primer and reverse primer (10 μmol/μL for primers), and double-distilled water to supply a final volume of 10 μL. The PCR was performed as follows: 95 °C for 2 min and 40 cycles of 95 °C for 10 s and 60 °C for 20 s. The expression level of each gene was summarized from three replicates for each sample. The house-keeping gene *β-actin* was used as an internal control for qRT-PCR and the relative expression levels were calculated using the 2^–^^ΔΔ^^Ct^ method [53].

### 2.4. Yeast Two-Hybrid Assay

The full-length coding sequences of *TeAP1-1*, *TeAP1-2*, *TeFUL1*, *TeFUL2*, and *TeFUL3* were amplified using primers with specific restriction sites and cloned into the activation domain plasmid pGBKT7 (Clontech, Palo Alto, CA, USA) and into binding domain plasmid pGADT7 (Clontech, Palo Alto, CA, USA), respectively. All constructs were confirmed by sequencing analyses. The primers were presented in Appendix A. The bait and prey constructs of five class B genes (*TeAP3-1*, *TeAP3-2*, *TePI*, *TeTM6-1*, and *TeTM6-2*) and class C + D genes (C: *TeAG1* and *TeAG2*, D: *TeAGL11-1* and *TeAGL11-2*) were previously described by Ai et al. [49] and Zhang et al. [50], respectively. The full-length sequences of six class E genes (*TeSEP1*, *TeSEP3-1*, *TeSEP3-2*, *TeSEP3-3*, *TeSEP4* and *TeAGL6*) were downloaded from NCBI, and bait and prey recombinants of these six class E genes were also constructed, respectively. Both bait and prey constructs were transformed into yeast cell strain *AH109* using LiAc method (Clontech) following the Frozen-EZ Yeast Transformation II Kit protocols (Zymo Research Corp, Irvine, CA, USA). Interaction results between bait proteins and empty AD, between prey proteins and empty BD, between empty BD and empty AD were used as negative controls. The interaction results between pGBKT7-53 and pGADT7-T7 vectors were used as a positive control. Yeast double transformants were plated onto SD medium without tryptophane (Trp) and leucine (Leu) (Sigma, St. Louis, MO, USA, A8056), and medium was incubated at 30 °C for 3-5 days. Positive clones were verified by PCR with general primers AD-R/F or BD-F/R (Appendix A). Three randomly selected positive yeast cells were spotted onto the X-α-gal-supplemented selection medium without Leu, Trp, histidine (His), and adenine (Ade). The interaction between the tested proteins was analyzed after 3–5-day incubation of the positive yeast cells at 30 °C.

### 2.5. Vector Construction and Plant Transformation

The full-length coding sequences of *TeAP1-1*, *TeAP1-2*, *TeFUL1*, *TeFUL2*, and *TeFUL3* were amplified by using primer pairs with specific restriction sites (Appendix A), and the amplification products were ligated to the *pCAMBIA2300s* plasmid, which harbored the CaMV35S promoter and kanamycin resistance (*Kan*) gene (Appendix A). The recombinant plasmids were named *35S:Te**AP1-1*, *35S:Te**AP1-2*, *35S:Te**FUL1*, *35S:TeFUL2*, and *35S:**TeAFUL3*, respectively. All the recombinant plasmids were introduced into *Escherichia coli DH5a* and tested by sequencing. These plasmids were separately transformed into chemically competent *Agrobacterium tumefaciens* strain *GV3101*, which was further transformed into wild-type Arabidopsis ecotype Columbia plants by the floral dip method [54]. T_1_ and T_2_ generation transgenic plants were selected in solid medium containing 50 μg/mL *kanamycin* and verified by PCR with a general forward primer of *35S*-F and gene-specific reverse primers *35S-TeAP1-1*-R, *35S-TeAP1-2*-R, *35S-TeFUL1*-R, *35S-TeFUL2*-R, and *35S-TeFUL3*-R (Appendix A), respectively. The genomic DNA was isolated from the transgenic plants and wild-type Arabidopsis, respectively. The transcript levels of *TeAP1-1*, *TeAP1-2*, *TeFUL1*, *TeFUL2*, and *TeFUL3* were analyzed by qRT-PCR and semi-quantitative PCR (Semi-PCR). The total RNA of blooming flowers from T_1_ transgenic plants and wild-type plants was isolated and reverse-transcribed with the above-mentioned reagent kit. The Arabidopsis house-keeping gene *EF1**α* (*AtEF1**α*, AT5G60390) was used as a control for qRT-PCR and semi-PCR. QRT-PCR were performed in the same way as described above, and the relative expression levels were calculated using the 2^−ΔΔCt^ method. The semi-PCR was performed as follows: 94 °C for 4 min, 24–26 cycles of 94 °C for 10 s, 60 °C for 30 s and 72 °C for 30 s, final extension for 5 min at 72 °C. The 24-26 cycles of semi-PCR were designed for the house-keeping gene *EF1α*, and 30–32 cycles of semi-PCR were designed for exogenous genes. Phenotype changes of T_1_ and T_2_ generation transgenic plants were analyzed. To testify the segregation tests, 16 kanamycin-resistant transgenic plants of the T_2_ generation lines that fitted a segregation ratio of 3:1 were chosen to record main morphological traits. The transcript levels of some endogenous genes of T_3_ generations were analyzed.

### 2.6. Expression Analysis of Endogenous Genes in Transgenic Plants

In order to investigate the conserved functions of *AP1/FUL*-like genes in marigold and to reveal the mechanism underlying phenotypic changes of transgenic lines *35S:TeAP1-2* and *35S:TeFUL2*, the transcript levels of some *AP1*-regulated endogenous genes (including *LFY*, *FT*, *SEP3*, *SOC1*, *SVP*, *TFL1*, *AGL24*, and *SPL9)* were analyzed by qRT-PCR [55]. Total RNA was isolated from 10-day-old T_3_ transgenic lines *35S:TeAP1-2* and *35S:TeFUL2* and wild-type Arabidopsis 10-day-old seedlings. Reverse transcription and qRT-PCR were performed in the same way as described above. The gene-specific primers are listed in Appendix A.

## 3. Results

### 3.1. Isolation and Phylogenetic Analysis of TeAP1/FUL-like Genes

The full-length sequences of five *AP1/FUL*-like genes were amplified by using gene-specific primers. In this study, the cDNA of ray floret sepals was used as a template to amplify the full-length sequences of *TeAP1**-**1* and *TeAP1-2*. The cDNA from different sizes of flower buds (0–1 mm in diameter and 3–4 mm in diameter) was used as a template to clone full-length sequences of *TeFUL1*, *TeFUL2*, and *TeFUL3*. In order to further identify the putative homologs of *AP1* and *FUL* genes, we blasted nucleotide sequences of these five genes against NCBI. The blast search results indicated that two different *AP1*-like genes and three different *FUL*-like genes were detected with the two *AP1*-like genes designated as *TeAP1-1* (Acc. No. MT394170), *TeAP1-2* (Acc. No. MT394171), and three *FUL*-like genes designated as *TeFUL1* (Acc. No. MT394172), *TeFUL2* (Acc. No. MT394173), and *TeFUL3* (Acc. No. MT394174), respectively. Sequence analysis revealed that the five putative proteins encoded by these five genes were composed of 246, 247, 235, 235, and 242 amino acids, respectively. The putative TeAP1-1 and TeAP2-1 proteins shared more than 89% amino-acid identity, and the identity between these two marigold putative proteins and one Arabidopsis AP1 clade protein was lower than 60% at the amino acid level (Appendix A). Three marigold putative FUL homologous proteins shared relatively low identity, and the identity between these three marigold putative FUL homologous proteins and two Arabidopsis FUL clade proteins was lower than 55% at amino acid level (Appendix A). Multiple sequence alignment and conservation analysis of AP1/FUL proteins indicated that all TeAP1/FUL proteins contained one conserved MADS domain, one less conserved I domain, one K domain, and one variable C-terminal domain (Figure 1). The putative proteins of TeAP1-1 and TeAP1-2 possessed one typical euAP1-motif (CFPS) containing both an acidic domain and a farnesylation motif (CaaX, shown at their C termini) (Figure 1). In addition, the characteristic FUL motif was shared by the three TeFUL proteins (Figure 1).

To investigate the relationship between *TeAP1/FUL* genes and other members of *AP1* and *FUL* clades, a phylogenetic analysis was carried out by using amino acid sequences of the AP1/FUL clade from other plant species and those of AGL6 subfamily proteins acting as an outgroup (Figure 2). TeAP1-1 and TeAP1-2 were orthologous to Arabidopsis AP1 and Antirrhinum SQUA. TeFUL1 and TeFUL3 were phylogenetically close to euFULI, and TeFUL2 was orthologous to the Antirrhinum protein DEFH28 belonging to euFULII protein. Notably, TeFUL1 and HaFUL (*H. annuus*) shared 78.39% amino-acid identity, and TeAP1-1 and HAM75 (*H*. *annuus*) were more closely related to each other with over 97.45% amino-acid identity. The high homology might indicate their functional similarity.

### 3.2. Expression Analysis of TeAP1/FUL-like Genes in Marigold

The expression patterns for these five *AP1/FUL* genes in different vegetative tissues, floral organs, and different development stages of flower buds were examined by qRT-PCR. *TeAP1-1* was mainly expressed in leaves, receptacles, bracts, sepals of ray florets, petals of disk florets, and ovaries but not expressed in flower buds and roots (Figure 3 and Appendix A). Compared to *TeAP1-1*, *TeAP1-2* was weakly expressed in different development stages of flower buds but was highly expressed in receptacles, sepals of ray and disk florets, petals of disk florets, and ovaries (Figure 3a and Appendix A). *TeFUL1* and *TeFUL3* shared a similar expression pattern, and they were widely expressed in vegetative and reproductive tissues (Figure 3a and Appendix A). Some differences in expression levels in some tissues were also detected between *TeFUL1* and *TeFUL3*. For example, *TeFUL1* was expressed mainly in petals of disk florets, stamens, ovaries and sepals of ray florets, while *TeFUL3* was highly expressed in all floral organs of two-type florets and receptacles (Figure 3b and Appendix A). Contrary to *TeFUL1* and *TeFUL3*, *TeFUL2* was highly expressed in vegetative tissues, flower buds, receptacles, and bracts, and it was weakly expressed in floral organs (Figure 3a and Appendix A).

### 3.3. Interactions between TeAP1/FUL Proteins and Other MADS-Box Proteins in Marigold

A yeast two-hybrid analysis was performed to evaluate the interaction strength between AP1/FUL proteins and class B, class C, class D, or class E proteins. The marigold proteins were individually fused to the binding domain and the activation domain and then were pairwise recombined in both directions. No autoactivation was observed among these five proteins (Appendix A). As shown in Table 1 and Appendix A, TeAP1/FUL proteins only interacted with SEPATELLA (SEP) proteins, but they did not interact with class B (TePI, TeAP3-1, TeAP3-2, TeTM6-1 and TeTM6-2), class C (TeAG1 and TeAG2), and class D (TeAGL11-1 and TeAGL11-2) proteins. TeAP1-1 and TeAP1-2 exhibited a similar protein interaction pattern, both of which interacted with class E proteins TeSEP3-2 and TeSEP3-3 to form heterodimers (Table 1, Appendix A). Contrary to two TeAP1 proteins, three TeFUL proteins displayed different protein interaction patterns. TeFUL1 only interacted with TeAGL6. TeFUL2 interacted with TeSEP3-2, TeAGL6, and itself. TeFUL3 interacted with TeSEP1, TeSEP3-1, TeSEP3-2, TeSEP3-3, and TeSEP4 to form heterodimers (Table 1, Appendix A).

### 3.4. Early Flowering Caused by Ectopic Expression of TeAP1-2 and TeFUL2 in Arabidopsis

To explore the potential functions of *TeAP1-1*, *TeAP1-2*, *TeFUL1*, *TeFUL2*, and *TeFUL3* genes, functional analyses were performed by overexpressing these five genes in Arabidopsis with the cauliflower mosaic virus *35S* promoter. After kanamycin selection and PCR verification, a total of 63, 31, 42, 26, and 45 independent T_1_ transgenic plants (namely, *35S:TeAP1-1*, *35S:TeAP1-2*, *35S:TeFUL1*, *35S:TeFUL2*, and *35S:TFUL3)* were obtained, respectively. The transcript level analysis revealed that five *AP1/FUL*-like genes (*TeAP1-1*, *TeAP1-2*, *TeFUL1*, *TeFUL2*, and *TeFUL3*) were successfully expressed in Arabidopsis plants (Figure 4l,m, and Appendix A). Compared with wild-type plants, the transgenic plants containing *35S:TeAP1-1*, *35S:TeFUL1*, and *35S:TeFUL3* exhibited no visual phenotypical changes. However, fourteen *35S:TeFUL1* and eleven *35S:TeFUL3* transgenic plants displayed early flowering. According to the phenotypic alterations, 2–6 T_2_ transgenic lines were, respectively, selected from the Arabidopsis transgenic plant whose progenies showed a 3:1 segregation ratio for kanamycin resistance, which may indicate a single-copy insertion of transgenes. Sixteen T_2_ transgenic plants for each line were used to investigate the flowering time and floral phenotypes.

Compared with the wild-type plants, overexpression of *TeAP1-2* and *TeFUL2* in Arabidopsis caused obvious early flowering (Figure 4d,e and Figure 5). According to the statistics, wild-type Arabidopsis flowered in ten to thirteen (11.17 ± 1.11) rosette leaves, while most transgenic lines flowered in five to eight rosette leaves under the same conditions (Figure 5). In addition, ectopic expression of *TeAP1-2* also led to the wavy shape of the last two rosette leaves and the curling of cauline leaves (Figure 4a–c). However, overexpression of *TeAP1-2* and *TeFUL2* in Arabidopsis could not affect flower development (Figure 4f–k).

### 3.5. Expression Analysis of Endogenous Genes in Transgenic Plants

To reveal the mechanism underlying phenotypic changes of transgenic lines *35S:TeAP1-2* and *35S:TeFUL2*, the expression levels of *AP1*-regulated endogenous genes were analyzed when the T_3_ transgenic and wild-type seedlings were 10 days old. As shown in Figure 6, *TeAP1-2* and *TeFUL2* displayed a similar function in regulating the expression level of *AP1* downstream genes. For example, the expressions of *AP1*, *FT*, *LFY*, *SOC1*, *SPE3*, and *TFL1* in transgenic lines *35S:TeAP1-2* and *35S:TeFUL2* were obviously higher than those in wild-type plants. The expression level of *AGL24* showed no remarkable changes in both transgenic lines *35S:AP1-2* and *35S:FUL2*. It should be noted that *TeAP1-2* and *TeFUL2* specifically regulated some downstream genes. For instance, *SPL9* was significantly upregulated in transgenic lines *35S:TeAP1-2*, while it exhibited no change in transgenic lines *35S:FUL2*. In contrast to *SPL9*, *SVP* was significantly highly expressed in transgenic lines *35S:FUL2*, but there was no significant change in transgenic lines *35S:AP1-2*.

## 4. Discussion

The study of many *AP1/FUL*-like genes from various species has demonstrated that *AP1/FUL* genes play key roles in flowering time, flower and fruit development. Like *APETALLA3* (*AP3*, B class gene) and *AGAMOUS* (*AG*, C class gene), the *AP1/FUL* genes underwent several duplication events, resulting in the occurrence of *euAP1* and *euFUL* clade in core eudicots [13,14]. In this study, five marigold *AP1/FUL*-like genes were obtained. Sequence alignment analysis indicated that all these five TeAP1/FUL-like proteins were typical MIKC proteins, and they contained a conserved motif at their C terminal domain (Figure 1). TeAP1-1 and TeAP1-2 were clustered into euAP1 clade proteins harboring an acidic domain and a farnesylation motif (Figure 1), and the TeFUL1, TeFUL2, and TeFUL3 possessed a conserved FUL motif (Figure 1), which suggested these three marigold FUL-like proteins were members of FUL clade proteins [13,14]. Such changes in the 3′ end of coding sequence have been explained by a frameshift mutation in ancestral *AP1/FUL*-like genes [13,56] and are responsible for gene-specific functions.

Our phylogenetic analysis indicated that TeAP1-1 and TeAP1-2 were members of the AP1 clade, and they seemed to be homologous to Antirrhinum SQUA, which was previously reported to be involved in regulating the floral meristem development and specifying the sepal and petal identity [57]. TeFUL1, TeFUL2, and TeFUL3 were clustered into the FUL clade, and TeFUL1 and TeFUL3 proteins were close to the euFULI group. TeFUL2 belonged to the euFULII group (Figure 2). *TeFUL2* was orthologous to the antirrhinum *DEFH28* involved in regulating floral meristem development, fruit development, and flowering time [40]. Gene expression analysis indicated that *TeFUL2* was mainly expressed at the early stage of inflorescence development (Figure 3 and Appendix A), and the expression pattern of *TeFUL2* was similar to that of the early function genes Arabidopsis *FUL* [23] and petunia *PFG* [42], implying a role of *TeFUL2* in meristem identity. However, *TeFUL1* and *TeFUL3* were expressed in vegetative tissues, different stages of flower buds, and floral organs (Figure 3 and Appendix A). Based on these findings, it could be speculated that *TeFUL2* and *TeFUL1* (or *TeFUL3*) might arise from gene duplication and that this duplication event might cause the change in their expression patterns. Many previous studies reveal that functional divergence is caused by gene duplication, which further drives evolution [10,58]. Therefore, we speculated that the duplication events and transcript pattern differences of *TeFUL* genes might imply the functional divergence of these genes in marigold.

### 4.1. Conserved Function of AP1/FUL Genes in Early Flowering

Functional analysis of the *AP1/FUL*-like genes in core eudicots and non-core eudicots reveal that *AP1/FUL*-like genes display conserved roles in regulating the flowering time. For example, overexpression of *AP1* or *FUL* in Arabidopsis both leads to early flowering [29,59]. Furthermore, a similar phenotype is also observed in the case of ectopic overexpression of *AP1*-like or *FUL*-like genes from the Asteraceae species, such as *C*. *morifolium* (*CDM111*) [45], *C*. *lavandulifolium* (*ClM8*) [44] and *G*. *hybrida* (*GSUQA2*) [43]. In this study, heterologous expression of *TeAP1-2* and *TeFUL2* in Arabidopsis resulted in early flowering without affecting floral organ identity (Figure 4d–k and Figure 5). In addition, ectopic expression of *TeAP1-2* also led to the wave-shaped rosette leaf and curled cauline leaf (Figure 4b,c), which was similar to the function of the *AP1/FUL*-like gene *MBP20* [60]. The MADS-box transcription factors possess a DNA-binding domain to regulate their downstream gene expressions [58]. Therefore, we speculated that the early flowering phenotypes observed in *35S:TeAP1-2* and *35S:TeFUL2* transgenic lines might be related to the change in endogenous gene expression levels. In this study, *AP1*, *FT*, *LFY*, *SOC1*, and *SEP3* were significantly upregulated in 10-day-old seedlings of transgenic lines containing *35S:TeAP1-2* and *35S:TeFUL2* fusion vectors (Figure 6), suggesting *TeAP1-2* and *TeFUL2* might share the overlapping regulation network of a series of downstream genes in Arabidopsis. Remarkably, the *TFL1* was significantly activated in transgenic seedlings overexpressing *TeAP1-2* or *TeFUL2* (Figure 6), which was consistent with the previous report that overexpression of the *FUL*-like gene *PlacFL2* from *Platanus acerifolia* Willd. obviously activated the *TFL1* expression [59]. However, in Arabidopsis, the *TFL1* inhibits *AP1* activities through transcriptional repression [55,61]. Therefore, our results require to be further investigated. In Arabidopsis, *AP1* directly represses *SVP*, *AGL24,* and *SOC1* to partially specify floral meristem identities [62]. However, in our study, no remarkable change in the expression level of *AGL24* was observed in transgenic lines *35S:TeAP1-2* and *35S:TeFUL2* (Figure 6). Additionally, the expression level of the flowering repressor gene *SVP* was significantly activated in transgenic lines *35S:TeFUL2*, but not in transgenic lines *35S:TeAP1-2* (Figure 6). In contrast to *SVP*, *SPL9* was significantly upregulated in transgenic lines *35S:TeAP1-2*, but not in transgenic lines *35S:TeFUL2* (Figure 6). These results revealed that *TeAP1-2* and *TeFUL2* had divergent functions in regulating downstream genes, which was further supported by their difference in protein interaction patterns (Table 1, Appendix A).

### 4.2. Potential Redundant Function of TeAP1-1 and TeAP1-2 as Class A Genes

In Arabidopsis, *AP1* is an early-acting gene, and it functions as a class A gene to specify sepal and petal identity [22,63]. *AP1* is expressed in floral meristems and developing sepal and petal primordia [22,23,26,64]. However, in other core eudicots, the *AP1*-like genes can also be expressed in bracts and reproductive organs [31,45,65,66]. Similarly, both *TeAP1-1* and *TeAP1-2* were highly expressed in sepals of two-type florets and petals of disk florets, bracts, receptacles, and ovaries (Figure 3 and Appendix A). Previous studies have revealed that the *AP1* gene is involved in the specification of floral meristem (FM) identity and its high expression in inflorescence meristems and inflorescence branch meristem of Cornus species tends to form closed inflorescences [22,25,67,68]. In this study, *TeAP1-2* was relatively highly expressed in flower buds, implying *TeAP1-2* might regulate the head flower formation. According to the floral quartet model, the combinations of class A and E proteins specify the sepal identity [5,69]. In this study, TeAP1-1 and TeAP1-2 shared a similar protein interaction pattern to form heterodimers with TeSEP3-2 and TeSEP3-3 (Table 1, Appendix A). In Arabidopsis, AP1 only interacted with SEP to form a heterodimer. Furthermore, in the Asteraceae species, the AP1-like proteins *C*. *morifolium* CDM111 [45,46], *G*. *hybrida* GSQUA1, and GSQUA3 [70] also had a limited protein interaction pattern. In other words, they only interacted with SEP proteins to form heterodimers. Additionally, the results of protein-protein interaction also imply that the TeSEP3 proteins played a glue role in regulating floral organ development. Taken together, as class A genes, *TeAP1-1* and *TeAP1-2* might play a redundant role.

### 4.3. Divergent Functions among TeFULs Genes

It is well-known that *FUL*-like genes play important roles in the transition from vegetative meristems to reproductive meristems and in fruit development in many core eudicots and non-core eudicots. In the model plant Arabidopsis, *FUL* regulates the cell differentiation during fruit development [37,39,61] and participates in specifying floral meristem identity together with *AP1* and *CAL* [23]. In basal eudicots, the *Aquilegia coerulea* Pall. *FUL*-like genes regulate leaf morphogenesis and inflorescence development [8]. Additionally, in monocots, *Oryza sativa* L. homologs genes *OsMADS14* and *OsMADS15* are involved in specifying the meristem identity, palea and lodicule identity [7]. In contrast to the *AP1*-like genes, the *FUL*-like genes are widely expressed in vegetative and reproductive tissues [6,37,59].

In our study, *TeFUL1* and *TeFUL3* were expressed in stems and leaves as well as in reproductive tissues (Figure 3 and Appendix A), which was in line with the typical *FUL*-like expression pattern [6,37,59], implying that *TeFUL1* and *TeFUL3* might play a role as *FUL* genes. Furthermore, ectopic expression of *TeFUL1* or *TeFUL3* in Arabidopsis led to no visible phenotype changes. In Arabidopsis, *FUL* functions redundantly with *CAL* and *AP1* to specify the floral meristem identity, and single *ful* mutation has no ability to affect floral organ identity [23]. In general, we speculated that *TeFUL1* and *TeFUL3* might function redundantly in regulating the floral meristem identity, or that *TeFUL1* and *TeFUL3* need to work together with *AP1*-like genes to regulate the floral meristem development. However, the striking difference in the protein interaction pattern was observed between TeFUL1 and TeFUL3 (Table 1, Appendix A). TeFUL1 only interacted with TeAGL6, while TeFUL3 interacted with TeSEP1, TeSEP3-1, TeSEP3-2, TeSEP3-3, and TeSEP4 to form heterodimers (Table 1, Appendix A). Different protein interaction patterns might be related to their different conserved regions at C domains (Figure 1). The above results suggested that *TeFUL1* and *TeFUL3* might be partially functionally redundant, but they might have their own specific functions in regulating floral organ identity.

In contrast to *TeFUL1* and *TeFUL3*, *TeFUL2* was highly expressed in flower buds and vegetative tissues, and weakly expressed or unexpressed in floral organs and ovules (Figure 3 and Appendix A). Additionally, TeFUL2 could form a homodimer by itself; meanwhile, it could form heterodimers with TeAGL6 and TeSEP3-2 (Table 1, Appendix A). Ectopic expression of *TeFUL2* in Arabidopsis also led to early flowering with fewer rosette leaves (Figure 5), which was consistent with the phenotype of the plants overexpressing *euFULII* (*DEFH28*) clade genes from core eudicots and non-core eudicots [34,43]. The above results suggested that *TeFUL1* and *TeFUL3* might lose some functions, but these functions might have been retained in *TeFUL2*. Overexpression of Antirrhinum *DEFH28* (*euFULII* clade genes) in Arabidopsis resulted in early flowering, two to four carpel formations, and failure of silique dehiscence [34]. However, ectopic expression of *TeFUL2* in Arabidopsis did not affect floral organ identity and silique dehiscence (Figure 4), which was in line with the study results of *G. hybrida GSQUA2* [43]. In general, *TeFUL2* might retain a conserved role in regulating the meristem transition rather than fruit ripping.

## 5. Conclusions

In conclusion, marigold has five *AP1/FUL*-like genes, two of which are clustered into the *euAP1* clade and three to the *FUL*-like clade. Based on the analyses of gene expression and protein interaction patterns, *TeAP1-1* and *TeAP1-2* are likely to play a redundant role in regulating sepal and petal identity, which is similar to the function of class A genes. Additionally, ectopic expression of *TeAP1-2* resulted in early flowering, implying that *TeAP1-2* might be involved in the regulation of floral transition. However, three *FUL*-like genes display divergent functions. *TeFUL1* and *TeFUL3* are more functionally close to *euFUL* genes, whereas *TeFUL2* is more functionally close to antirrhinum *DEFH28* belonging to the *euFULII* gene. Our results will provide a theoretical basis for the study of class A genes in Asteraceae. Considering the great difference in the florescence structure between marigold and Arabidopsis, this study will be helpful for understanding the function of *AP1/FUL* genes in Asteraceae species.

## Figures and Tables

**Figure 1 genes-12-02011-f001:**
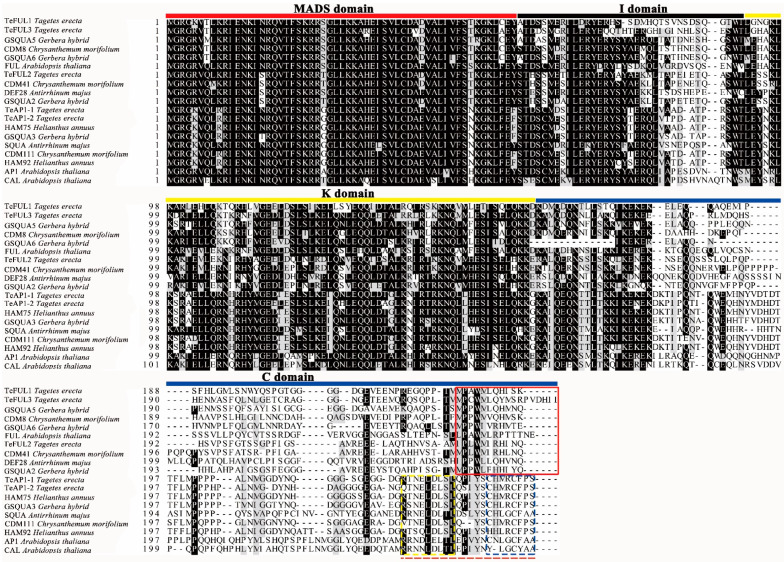
Alignment of marigold AP1/FUL-like amino acid sequence from model plants (Arabidopsis and Antirrhinum) and Asteraceae species. The MADS domain is marked with a bold red line. The I domain is marked with a bold black line. The K domain is marked with a bold yellow line. The C domain is marked with a bold blue line. The FUL protein motif is marked with a red box. EuAP1-like proteins contain both an acidic domain (shown in yellow dotted box), and a farnesylation motif (shown in blue dotted box) at their C termini. The whole domain marked with a red dotted line represents the euAP1 motif.

**Figure 2 genes-12-02011-f002:**
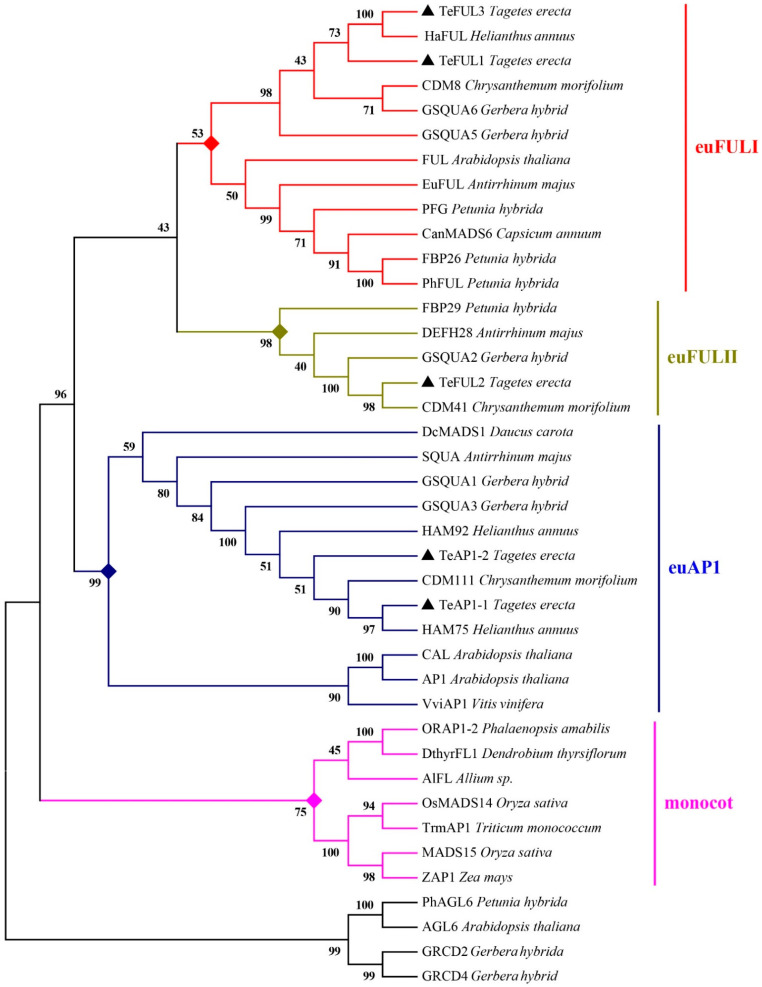
Phylogenetic tree based on the amino-acid alignment of TeAP1/FUL proteins. The tree was generated with the MEGA v7.0 software, using the neighbor-joining (NJ) method and 1000 bootstrap replicates. The TeAP1-1, TeAP1-2, TeFUL1, TeFUL2, and TeFUL3 are marked with black triangles.

**Figure 3 genes-12-02011-f003:**
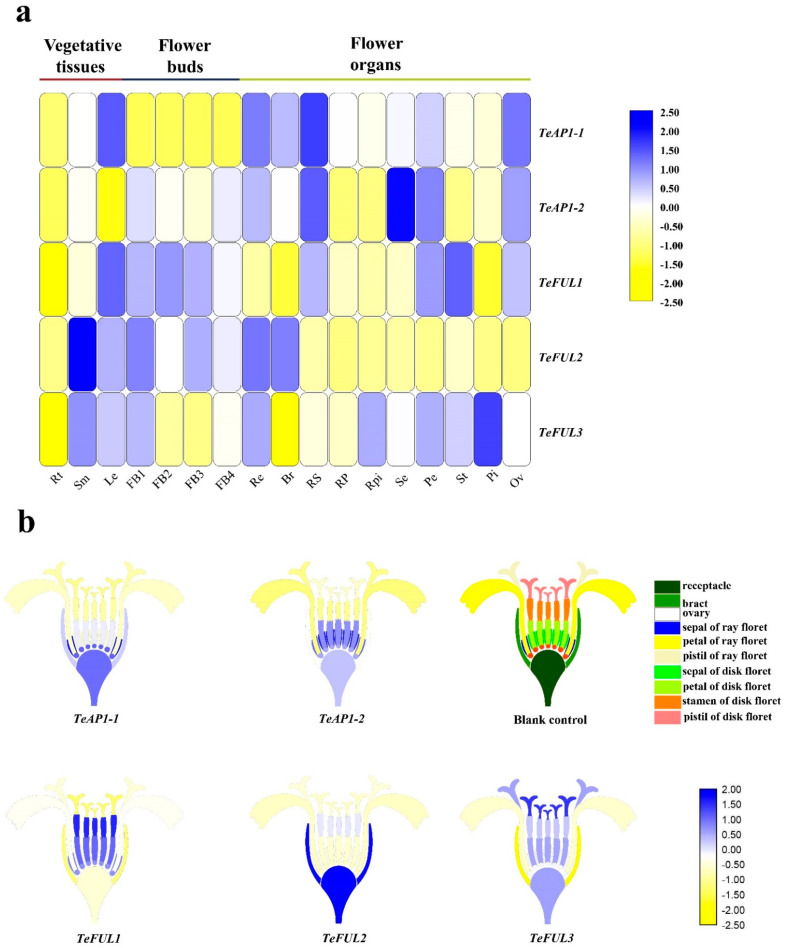
Expression levels of *TeAP1/FUL* genes in different tissues and organs of marigold. (**a**) Heatmap of relative expression of *TeAP1/FUL* genes by qRT-PCR in different tissues and organs. Rt: root; Sm: stems; Le: leaves; FB1-FB4: flower buds were 0–1, 2–3, 4–5 and 6–7 mm in diameter, respectively; Re: receptacle; Br: bract; RS: sepal of ray floret; RP: petal of ray floret; RPi: pistil of ray floret; Se: sepal of disk floret; Pe: petal of disk floret; St: stamen of disk floret; Pi: pistil of disk floret; Ov: ovary. (**b**) Heatmap of *TeAP1/FUL* genes in the inflorescence of marigold based on the relative expression by qRT-PCR. Blank control: structural model of capitulum in marigold; different colors represent different floral organs.

**Figure 4 genes-12-02011-f004:**
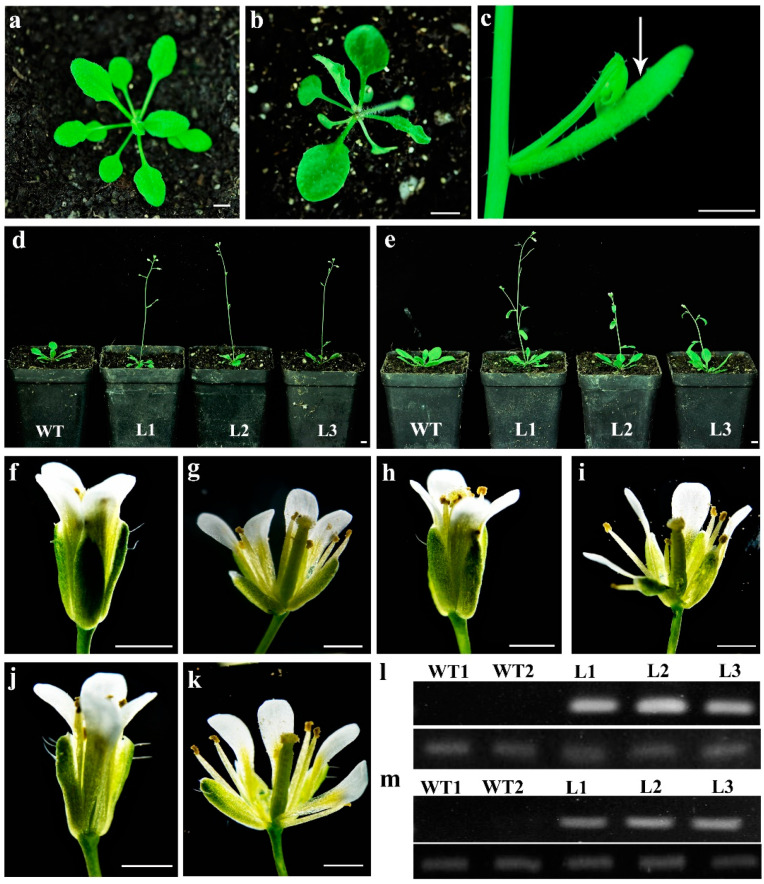
Abnormal morphology of transgenic Arabidopsis plants of constitutively expressed *TeAP1-2* and *TeFUL2* genes. (**a**) The wild-type seedling. (**b**) The transgenic seedlings with severely curled rosette leave in *35S:TeAP1-2* transgenic lines. (**c**) The curled cauline leaves of *35S:TeAP1-2* transgenic lines. (**d**) Wild-type (left) and early flowering transgenic plant (right) of *35S:TeAP1-2* transgenic lines; WT: wild-type line 1; L1: *35S:TeAP1-2* line 1; L2: *35S:TeAP1-2* line 2; L3: *35S:TeAP1-2* line 3. (**e**) Wild-type (left) and early flowering transgenic plant (right) of *35S:TeFUL2* transgenic lines; WT: wild-type line 1; L1: *35S:TeFUL2* line 1; L2: *35S:TeFUL2* line 2; L3: *5S:TeFUL2* line 3. (**f**) The flower of wild-type Arabidopsis plants. (**g**) The anatomy of wild-type Arabidopsis flower. (**h**) The flower of *35S:TeAP1-2* transgenic plants. (**i**) The anatomy of *35S:TeAP1-2* transgenic plant flowers. (**j**) The flower of *35S:TeFUL2* transgenic plants. (**k**) The anatomy of *35S:TeFUL2* transgenic plant flowers. (**a**–**e**), bar = 5 mm; (**f**–**k**), bar = 500 μm. (**l**) Expression of *TeAP1-2* in seedlings of T_1_ transgenic plants by semi-RT-PCR. The picture above is the expression level of *TeAP1-2* in transgenic lines, the band size was 106 bp; the picture below the constitutive gene is the Arabidopsis keeping-house gene *AtEF1α*. (**m**) Expression of *TeFUL2* in seedlings of T_1_ transgenic plants by semi-RT-PCR. The picture above is the expression level of *TeFUL2* in transgenic lines, the band size was 127 bp; the picture below the constitutive gene is Arabidopsis house-keeping gene *AtEF1α*.

**Figure 5 genes-12-02011-f005:**
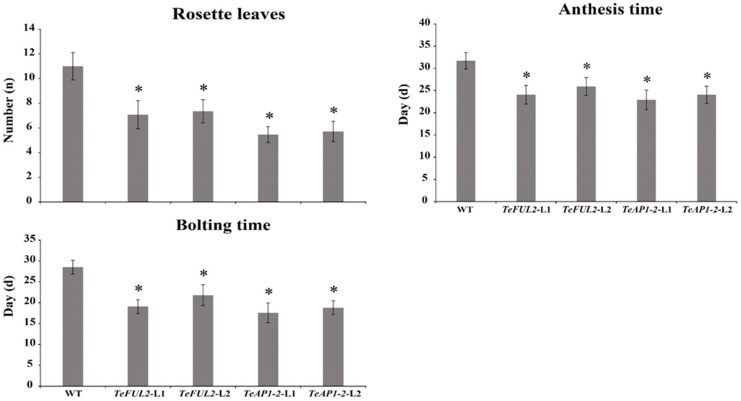
Statistics for main morphological traits of the control and transgenic plants. * Significant difference at *p* < 0.05.

**Figure 6 genes-12-02011-f006:**
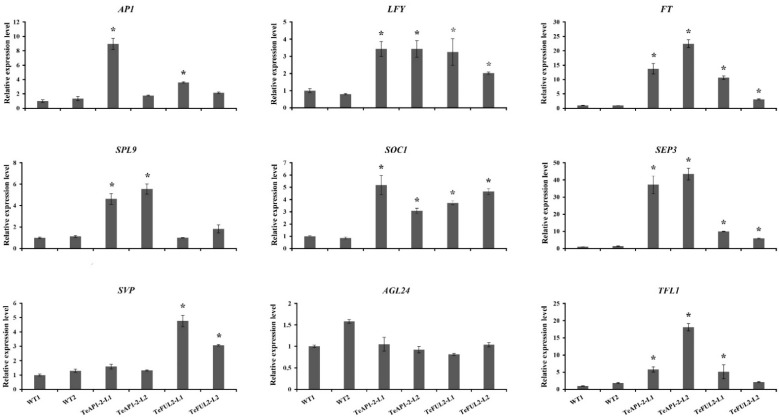
qRT-PCR analysis of endogenous flowering related genes in 10-D-old seedlings of Arabidopsis wild-type and *35S:TeAP1-2* and *35S:TeFUL2* transgenic lines. WT1: wild-type line 1; WT2: wild-type line 2; * expression level of endogenous genes in transgenic plants was 2 times higher or 0.5 times lower than that in wild-type plants.

**Table 1 genes-12-02011-t001:** Interactions of Marigold TeAP1/FUL proteins with classes A, B, C, D, and E proteins detected by yeast two-hybrid assays.

AD BD	Class A Proteins					Class B Proteins					Class C Proteins		Class D Proteins		Class E Proteins					
	TeAP1-1	TeAP1-2	TeFUL1	TeFUL2	TeFUL3	TePI	TeAP3-1	TeAP3-2	TeTM6-1	TeTM6-2	TeAG1	TeAG2	TeAGL11-1	TeAGL11-2	TeSEP1	TeSEP3-1	TeSEP3-2	TeSEP3-3	TeSEP4	TeAGL6
TeAP1-1	-	-	-	-	-	-	-	-	-	-	-	-	-	-	-	-	++	+	-	-
TeAP1-2	-	-	-	-	-	-	-	-	-	-	-	-	-	-	-	-	+	+	-	-
TeFUL1	-	-	-	-	-	-	-	-	-	-	-	-	-	-	-	-	-	-	-	+
TeFUL2	-	-	-	+	-	-	-	-	-	-	-	-	-	-	-	-	-	-	-	+
TeFUL3	-	-	-	-	-	-	-	-	-	-	-	-	-	-	++	+	++	++	-	-
TePI	-	-	-	-	-	/	/	/	/	/	/	/	/	/	/	/	/	/	/	/
TeAP3-1	-	-	-	-	-	/	/	/	/	/	/	/	/	/	/	/	/	/	/	/
TeAP3-2	-	-	-	-	-	/	/	/	/	/	/	/	/	/	/	/	/	/	/	/
TeTM6-1	-	-	-	-	-	/	/	/	/	/	/	/	/	/	/	/	/	/	/	/
TeTM6-2	-	-	-	-	-	/	/	/	/	/	/	/	/	/	/	/	/	/	/	/
TeAG1	-	-	-	-	-	/	/	/	/	/	/	/	/	/	/	/	/	/	/	/
TeAG2	-	-	-	-	-	/	/	/	/	/	/	/	/	/	/	/	/	/	/	/
TeAGL11-1	-	-	-	-	-	/	/	/	/	/	/	/	/	/	/	/	/	/	/	/
TeAGL11-2	-	-	-	-	-	/	/	/	/	/	/	/	/	/	/	/	/	/	/	/
TeSEP1	-	-	-	-	-	/	/	/	/	/	/	/	/	/	/	/	/	/	/	/
TeSEP3-1	-	-	-	-	+	/	/	/	/	/	/	/	/	/	/	/	/	/	/	/
TeSEP3-2	+	-	-	+	+	/	/	/	/	/	/	/	/	/	/	/	/	/	/	/
TeSEP3-3	-	-	-	-	+	/	/	/	/	/	/	/	/	/	/	/	/	/	/	/
TeSEP4	-	-	-	-	++	/	/	/	/	/	/	/	/	/	/	/	/	/	/	/
TeAGL6	-	-	-	-	-	/	/	/	/	/	/	/	/	/	/	/	/	/	/	/

Note: ++, strong interaction; +, weak interaction; -, no interaction; /, not tested.

## Data Availability

The data presented in this study are available on request from the corresponding author. The data are not publicly available due to privacy considerations.

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
