# Peer review of "Functional Conservation and Divergence of Five AP1/FUL-like Genes in Marigold (Tagetes erecta L.)"

_genes, 2021, doi:10.3390/genes12122011_

Round 1

Reviewer 1 Report

Authors presented an interesting research related to cloning and tissue distribution of five AP1/FUL-like genes in marigold. Phylogenetic studies, Y2H test and gene overexpression in A. thaliana, enabled to find putative roles of these genes in floral development.

Some minor comments are presented below:

Section 2.2

Details of PCR stages used to amplify cDNA of cloned genes.

Section 2.3

How the RNA was isolated?

Details of DNase treatment to remove putative remnants of genomic DNA

Reverse transcription reaction; details of reaction-temperature and time, volume and amount of used RNA

qPCR target- length of PCR product (control beta-actin and tested genes), target gene symbol and accession number.

qPCR protocol: conditions of PCR reaction, volume of reaction, concentration of magnesium ions, dNTPs, DNA polymerase type and concentration.

Software used to perform and analyse RT-PCR results.

Section 2.6

Concentration of kanamycin applied to select transformants.

Other:

Figure 5- should be Rosette leaves not Rosettle leaves

Author Response

Some minor comments are presented below:

Section 2.2

Details of PCR stages used to amplify cDNA of cloned genes.

Response:

As you suggest, we have added more detail for PCR stages in section 2.2, please see the revised manuscript for details (e.g. method section 2.2, line 152-154).

Section 2.3

How the RNA was isolated?

Response:

In this study, the total RNA was isolated with PLANTpure kit (Aidlab, Beijing, China) according to the manufacturer’s protocol.

As you suggest, we have added more details for this section, please see the revised manuscript for detail (e.g. method 2.3 section, line 174-178).

Details of DNase treatment to remove putative remnants of genomic DNA

Response:

In this study, the remnants of genomic DNA from total RNA was removed by the TRUEscript RT reagent Kit with gDNA Eraser (Aidlab, Beijing, China) with the Oli-go-dT primers. Here, we have added more details for this section, please see the revised manuscript for detail (e.g. method 2.3 section, line 179-183).

Reverse transcription reaction; details of reaction-temperature and time, volume and amount of used RNA

Response:

The details of reverse transcription reaction; details of reaction-temperature and time, volume and amount of used RNA were provided in revised manuscript (e.g. method 2.3 section, line 179-183).

qPCR target- length of PCR product (control beta-actin and tested genes), target gene symbol and accession number.

Response:

The qPCR target- length of PCR product and target gene symbol were added in revised manuscript (e.g. method 2.3 section, line 185-188).

The accession number were listed in results 3.1 section (e.g. Results 3.1 section, line 273-275).

qPCR protocol: conditions of PCR reaction, volume of reaction, concentration of magnesium ions, dNTPs, DNA polymerase type and concentration.

Response:

In this study, the qPCR was carried out using SYBR Primix Ex Taq kit (TaKaRa, Dalian, China). The PCR reaction, volume of reaction, concentration of magnesium ions, dNTPs, DNA polymerase type and concentration has been described in detail, please see the detail in method section 2.3, line 193-197.

Software used to perform and analyse RT-PCR results.

Response:

The gene expression levels were calculated by ABI Prism 7500 Sequence Detection System Software (Applied Biosystems, USA). The RT-PCR results were calculated using the 2– ΔΔCt method.

Section 2.6

Concentration of kanamycin applied to select transformants.

Response:

In this study, the transformants were selected on a solid medium containing 50 μg/ml kanamycin. We have added more details for this section, please see the revised manuscript for detail (e.g. method section 2.5, line 235-236).

Other:

Figure 5- should be Rosette leaves not Rosettle leaves

Response:

As you suggest, we have corrected it in Figure 5, please see the revised Figure 5 for detail.

Reviewer 2 Report

Now genetics is leaving through a new stage in its development – evaluation of genes’ evolution (including development of their structure and functions). The presented article is devoted to the testing of structure and functions of marigold APETALA1/FRUITFULL genes, which play an important role in plants development. Their structure and functions were evaluated in model plants. It was discovered that they evolved through duplications and mutagenic diversification. It is very interesting from the theoretical point of view for the understanding of molecular mechanisms of evolution. In marigold authors discovered five AP1/FUL-like genes. Two of them form euAP1 clade, and the rest compose FUL-like clade. Also it was supposed that TeAP1-1 and TeAP1-2 are are involved in the regulation of sepals and petals identity. Also, ectopic expression of TeAP1-2 led to early flowering, showing that TeAP1- 2 might be involved in the regulation of transition to flowering. So, it evolution of duplicated genes resulted in formation of new functions.

My only remark is that it would be correct to add the names of botanical species authors:

Tagetes erecta L., Arabidopsis thaliana L., etc.

Author Response

My only remark is that it would be correct to add the names of botanical species authors:

Tagetes erecta L., Arabidopsis thaliana L., etc. y only remark is that it would be correct to add the names of botanical species authors:

Tagetes erecta L., Arabidopsis thaliana L., etc.

Response:

Thank you for your suggestion. As you suggest, we have corrected it in revised manuscript, please see the revised manuscript for details (line 3, 13, 33-34, 61-63, 94-95, 107, 112, 129, 465, 505, and 507).